

# Attitudes towards Italian Mafias Scale (AIMS): development and validation

Michael Schepisi[1,2], Marco Tullio Liuzza[3], Althea Frisanco[1,2],
Anna Maria Giannini[4] and Salvatore Maria Aglioti[1,2]

[1] Department of Psychology, Sapienza University of Rome Center for Life Nano-&Neuroscience, Fondazione Istituto Italiano di Tecnologia (IIT), Rome, Italy
[2] IRCSS Santa Lucia Foundation, Rome, Italy
[3] Department of Medical and Surgical Sciences, Magna Graecia University of Catanzaro, Catanzaro, Italy
[4] Department of Psychology, University of Roma "La Sapienza", Rome, Italy

## ABSTRACT

In two studies we aimed at developing the Attitude towards Italian Mafias Scale (AIMS). In study 1 ($N = 292$) we used an Exploratory Factor Analysis to reduce the number of the items and explore their latent constructs. In study 2 ($N = 393$) we performed a Confirmatory Factor Analysis on the resulting 18-item questionnaire, whose latent structure was best identified by a general factor Mafia Attitude and three specific factors related to Behaviors, Cognitions and Emotions-Cognitions towards mafias. Moreover, we showed that the AIMS has (i) discriminant validity compared to a measure of attitudes towards crime, (ii) predictive validity of donation behavior to an association against mafias, (iii) internal consistency, and (iv) invariance for people of the five deep-rooted mafia regions of Southern Italy and those from the rest of Italy. Finally, we observed a difference between the participants from the five deep-rooted mafia regions (*i.e.*, Apulia, Basilicata, Calabria, Campania, Sicily) and the rest of Italy, with the former having surprisingly more negative attitudes towards mafias compared to the latter. The AIMS might help to reliably survey people's sentiment towards Italian mafias and promote targeted and effective law-related education interventions.

## INTRODUCTION

According to Article 46 *bis* of the Italian Criminal Code (the "La Torre law", 1982; *Republic of Italy, 1982*) mafias are organizations "of three or more people who make use of the power of intimidation afforded by the associative bond and the state of subjugation and criminal silence (corresponding to the Italian term *omertà*) which derives from such a bond to commit crimes, to acquire directly or indirectly the management or control of economic activities, concessions, authorizations or public contracts and services, to gain unjust profits or advantages for themselves or for other to prevent or obstruct the free exercise of the vote and to obtain votes for themselves or others during elections".

Although historically linked to Sicily, the term *mafia* can be applied to criminal organizations that developed in Southern Italy and that share the characteristics outlined above.

Corresponding author
Michael Schepisi,
michael.schepisi89@gmail.com

We can identify four mafia-like organizations, each developed in a specific region of Southern Italy, and named: Cosa Nostra in Sicily, 'ndrangheta in Calabria, Camorra in Campania, and Sacra Corona Unita in Apulia. In a fifth southern region, Basilicata, a less powerful and more recent mafia is represented by a network of clans called Basilischi (*Pantaleone, 2003*). Despite being strictly linked to other mafias from geographically contiguous regions (*i.e.,* 'ndrangheta from Calabria and Camorra from Campania), the Basilischi have been judicially acknowledged as a mafia (2007 and 2012 sentences by the Court of Potenza). Thus, here, we included this mafia too in what we refer to as deep-rooted mafia regions (DRMR).

Because of its pervasiveness and strong control over a given territory, where it competes with the State (*Sciarrone, 2009*), the mafia phenomenon has an extraordinary impact at economic, social and political levels (*Allum, Merlino & Colletti, 2019*). For example, the presence of mafia has dramatic social and economic costs for the above mentioned regions and for Italy more broadly: in the period from 2013 to 2017 mafias have been responsible for the 9% of the homicides committed over the national territory (https://www.istat.it/it/files/2018/11/Report_Vittime-omicidi.pdf), about half of non-domestic/non-intimate homicides, and more than 70% of the homicides related to systemic violence (*Aziani, 2022*). Moreover, mafia-like organizations gained 15.9 billion euros in 2015 only (*Savona & Riccardi, 2015*), and their presence in two of the deep-rooted mafia regions hampered the potential economic development of 16% GDP per capita (*Pinotti, 2015*).

Crucially, despite the increasing number of associations and initiatives against them (*Cinotti, 2015*), in the last decades mafias have been spreading outside their territories of origin, involving northern Italian regions (*e.g.*, Lombardy, Piedmont), as well as European (*e.g.*, Germany, The Netherlands) and extra-EU Countries (*e.g.*, Canada, Australia) (*Calderoni et al., 2016*). According to a report of 2014, just the 'ndrangheta counts as many as 60,000 associates spread in 30 countries all over the World (https://demoskopika.it/wp-content/uploads/2021/03/CS-Ndrangheta-Spa.pdf).

Traditionally, mafias mainly made use of violence, intimidation and subjugation to penetrate into and control their territories. However, in the last 25 years, after the strong reaction of the State to the murders of judges Giovanni Falcone and Paolo Borsellino, mafias realized that it was more advantageous to establish mutually useful relationships with the population, especially professionals whose skills can be exploited to make profits (*Sciarrone, 2014*; *Allum, Merlino & Colletti, 2019*). The collaborative degree of these relationships can go from mere "complicity" (a una tantum economic transaction) to "collusion" (a continuous exchange), and even reach a sort of "co-penetration", whereby the people involved (but un-affiliated to mafias) identify with and act as members of the clan (*Sciarrone, 2011*). Another way through which mafia-like organizations can exert their control is by appropriating the cultural values of the territories where they are present, and then convince the population that they represent those values (*Travaglino & Abrams, 2019*). In doing so, they can gain legitimization, support and, importantly for the present research, elicit favorable attitudes (*Travaglino & Abrams, 2019*). In fact, as posited by the seminal work by *Ajzen & Fishbein (1977)*, one fundamental factor that explains people's

behavior is their attitude, *i.e.,* a favorable or unfavorable evaluation towards the object of that behavior (*Eagly & Chaiken, 1993*). Thus, we argue that the thriving of mafia-like organizations might be eased by the positive attitude expressed by part of the population.

## Attitudes and their measures towards mafia, criminal organizations and crime

Along the decades scholars have proposed different models regarding attitudes structure: for instance, the "Multi-attribute Measurement Model" by *Fishbein (1963)* views an attitude as the summation between a certain belief of the attributes of the attitudinal object and the evaluation of those attributes. In the "Vector Model" by *Calder & Lutz (1972)* attitudes occupy a two-dimensional space made of an affective and a cognitive component. Extending these two-component models, the most prominent framework of attitudes structure comprises tripartite models, whereby attitudes consist of an affective, a cognitive, and a behavioral component (*Rosenberg et al., 1960*). The most known of these models is the ABC model by Eagly & Chaiken, for which the affective component refers to feelings and emotions, the cognitive component refers to beliefs, and the behavioral component to behavioral intentions towards the attitudinal object (*Eagly & Chaiken, 1998*; *Eagly & Chaiken, 2007*).

In many research fields attitudes have been measured through qualitative as well as quantitative methods, and research on crime and mafia make no exception (*e.g., Iacolino, Pellerone & Ferraro, 2017*; *Sarno, 2014*; *Travaglino, Abrams & De Moura, 2016*). Surveys and questionnaires might be particularly useful as they are easy to administer to large numbers of people, straightforward to understand, and give standardized results. Unfortunately, due to its contextual specificity, research and tools for measuring people's attitudes towards Italian mafias have been lacking. *Travaglino et al. (2014)* and *Travaglino, Abrams & De Moura (2016)* used a set of items to measure the mediating role of the attitudes towards Camorra (the Campania regional variation of mafia) in the relation between masculine and honor ideology and collective actions against Camorra itself. Although their measure displayed a good reliability, it has not been subjected to psychometric validation, and thus may not turn out to be the best tool for measuring attitudes towards mafias. In a similar vein, different associations against mafias periodically administer useful and relevant surveys but without submitting them to a proper validation procedure (see, for instance, *Della Ratta-Rinaldi, Ioppolo & Ricotta, 2012*).

Measurements of the attitudes towards other illegal phenomena (*e.g.,* drinking and driving; *Sprang, 1997*), anti-social behaviors, and attitudes towards crime in general are somewhat more reliable, as they underwent to proper systematization and psychometric validation. One of the first and widely employed measures is the "Criminal Sentiment Scale" (CSS; *Gendreau et al., 1979*), which consists of five subscales assessing the attitudes towards different crime-related entities (*i.e.,* law, court, police, tolerance towards violations and identification with criminal others). In a similar vein, the "Measure of Criminal Attitudes and Associates" (MCAA; *Mills, Kroner & Forth, 2002*) is a two-part questionnaire that asks people to indicate whether and to what extent their important others are criminals (part 1), and a series of attitudes towards criminal other themselves, violence, entitlement (*i.e., a*

person's sentiment to obtain whatever they want) and anti-social intent (part 2). A more general approach is instead employed in the "Attitudes towards Crime Scale" (ACS; *Ortet-Fabregat & Pérez, 1992*), which measures the attitudes towards three distinct dimensions, namely the (i) causes, (ii) prevention and (iii) treatment of crime. It is worth mentioning also the "Psychological Inventory of Criminal Thinking Styles" (PICTS; *Walters, 2007*), which assesses the eight thinking styles that are thought to support and reinforce the four behavioral patterns of criminal activity specified by the "Lifestyle Criminality Screening Form" (LCSF; *Walters, White & Denney, 1991*). However, none of the above instruments are tailored for the specific mafia phenomenon, which, due to the involvement of cultural, social, and political elements, differs from "classic" criminality. Moreover, although spread in different geographical contexts, mafia is a typical Italian phenomenon, which needs to be dealt with primarily within its context of origin.

With this in mind, this research aims at filling this gap by developing a reliable and psychometrically valid instrument for measuring the attitudes towards Italian mafias (AIMS). Such a tool could help researchers and policymakers to grasp the sentiment of the Italian population and promote ad hoc interventions to reduce the support for mafia-like organizations and their extremely negative societal consequences. To achieve this objective we conducted two studies, each with specific aims:

in Study 1 we aimed at reducing the number of the items generated in a previous stage and exploring the latent structure of the scale; in Study 2 the main purpose was to confirm the scale's structure explored in Study 1. In addition, we aimed at cross-validating the scale by testing (i) its discriminant validity against the revised version of the Criminal Sentiment Scale (CSS; *Gendreau et al., 1979*; *Shields & Simourd, 1991*); (ii) its criterion predictive validity in relation to donating real money to an association against mafias; (iii) its measurement invariance in participants born in the five deep-rooted mafia regions (DRMR; *i.e.,* Apulia, Basilicata, Calabria, Campania, and Sicily) and participants born in the rest of Italy.

## MATERIALS AND METHODS

### Items generation

To create the items of the AIMS we used a deductive method, also known as "classification from above" (*Hunt, 1991* in *Boateng et al., 2018*), whereby we reviewed the existing literature on Italian mafias in different research fields (*e.g.,* psychology, sociology, investigative journalism, law, history, *etc.*). Moreover, we asked two psychologists working for the Police and with experience with mafia criminals to review the content of the items. As the point of reference in research on attitudes, we developed our scale based on the framework comprising tripartite models of attitudes (*Eagly & Chaiken, 1998*; *Eagly & Chaiken, 2007*; *Rosenberg et al., 1960*). Thus, we created items referring to the emotional (*i.e.,* emotions), cognitive (*i.e.,* thoughts and beliefs), and behavioral (*i.e.,* behavioral intentions) components.

The initial pool of 76 items consisted of at least twice the number of items of the expected scale (see Supplementary materials for the complete list) in accordance with the suggestion by *Schinka, Velicer & Weiner (2012)*.

## Study 1

### *Participants*

We recruited 292 Italian participants (190 females; $M_{age} = 30.90$ years old, $SD_{age} = 11.36$) from a convenience sample collected through a snowball procedure. Fifty-five participants (18.8%) were born in Apulia, Basilicata, Calabria, Campania and Sicily, (*i.e.,* the five regions classically associated to the presence of mafia-like criminal organizations), the other 237 (81.2%) in the rest of Italy. Three respondents (1%) had a middle school diploma, 60 (20.5%) a high school diploma, 174 (59.6%) a bachelor's or master's degree, 36 (12.3%) a doctoral degree or a master post-lauream, and 6 (2.1%) had other diplomas. On a scale ranging from 0 to 100 participants' mean self-reported socio-economic status was 48.81 (SD = 17.22).

### *Measures*

Attitude towards Italian Mafias Scale (AIMS): the scale consisted of 76 items to which participants were asked to express their disagreement or agreement on a response scale ranging from 1 (completely disagree) to 7 (completely agree).

    Demographic: we collected participants' age, gender, region of birth, education, and socio-economic status.

### *Procedure*

The study was implemented in Qualtrics (Provo, UT), to which participants could grant access by clicking on the online link spread on the social media. Once they got access to the study, participants read the informed consent and clicked on a button to accept its terms. Then, they had to fill in the demographic questions and the 76 items of the AIMS. The order of presentation of the items of the AIMS was random.

    The procedure took approximately 40 min. Participants were not compensated for their involvement in the study.

    The present research (Prot. n. 0002192) was approved by the Sapienza University of Rome ethics committee and was conducted in accordance with the 1964 Declaration of Helsinki.

## RESULTS

### Items reduction

To reduce the high number of items we followed both a theory-driven and a data-driven approach. In the former, the decision to remove or retain a certain item was based on a thorough examination of its content and theoretical significance for the scale. In the latter, items that correlated less than |.2| with the total score (obtained by averaging all items of the scale) were considered for removal. In fact, an item that correlates less than |.3| or |.2| with the total score might be measuring something different compared to the other items in the scale (*Field, 2005*). Here, aware of the potential consequences of including items whose content might be very far from the construct of interest, we opted for the more conservative criterion of |.2|. Moreover, we excluded the items whose median scores corresponded exactly to the extremes of the response scale (*i.e.,* 1 or 7). In

fact, extreme response style (ERS) might affect the validity of the construct measurement (*Moors, 2004* in *Batchelor & Florida, 2016*). Items with such extreme values were also those that might have had the least discriminant power. Thus, we started by eliminating these items, then we eliminated one item (E3-"How much warm/cold do you feel with respect to mafia organized crime?") because, upon closer inspection, its labels were ambiguous: for instance, "warm" might have been interpreted both in terms of positive (*e.g.*, passion) and negative (*e.g.*, anger) emotions. We eliminated two items (E10, C32) because we realized that they did not precisely measure the attitudes towards mafias. We eliminated three items (B6, C31, C33) because they correlated indeed less than |.2| with the total score of the questionnaire. However, we retained five items (B22, B23, C8, C23, C25) because, even if poorly correlated with the total score, they were deemed as theoretically important.

The final pool consisted of 28 items (see Table 1).

### *Exploratory factor analysis*
The Kaiser–Meyer–Olkin test (Overall KMO = .76; all items KMOs > .5) showed that our sample size was adequate to perform exploratory factor analysis. Moreover, Bartlett's test of sphericity ($\chi^2(378) = 1876.05$, $p < .001$) indicated that the inter-items correlations were large enough, and Determinant > 0.00001 (det = .001) indicated no multicollinearity issues.

Given the ordinal nature of our variables, we performed an exploratory factor analysis (EFA) using the Principal Axis (PA) factor extraction method. Although parallel analysis suggested a 6-factor solution, we followed a simpler solution based on Kaiser's criterion of eigenvalues > 1 (*Kaiser, 1960*) and Cattel's criterion based on the scree-plot (*Cattell, 1966*), which both suggested a 3-factor solution explaining 26% of the total variance. We treated missing values with a listwise deletion procedure. Then, we conducted an EFA forced with a 3-factor solution. We used *oblimin* rotation to increase the interpretability of the factor solution, and because the underlying latent variables were likely to be correlated, which was indeed the case ($r_s < .36$). We excluded five items (C7, C18, C24, C27, C37) because either loaded poorly (r < .2) on a factor, or saturated equally on two or more of the factors (the difference between the item's highest and lowest loadings on the factors is r < .2), or -instead of focusing on the mafias per se- their content was more related to the attitudes towards the State (see Table 1 for factor loadings).

The final pool consisted of 23 items clustered around three factors: one factor gathered those items that referred to behavioral intentions towards mafias (*Behaviors*); one gathered those items that referred to beliefs and thoughts about mafias (*Cognitions*); and one gathered those items that referred to a mix of emotions and thoughts towards mafias (*Emotions-Cognitions*).

## Study 2
### *Participants*
We recruited 393 Italian participants (194 females; $M_{age} = 27.74$ years old, $SD_{age} = 8.25$) through Prolific (http://www.prolific.co). Among them, 198 (50.6%) were born in the five DRMR regions, while 193 (49.6%) in the rest of Italy. Four respondents (2%) had a middle school diploma, 133 (33.8%) a high school diploma, 222 (56.5%) a bachelor's

**Table 1  Study 1 items' factor loadings after Exploratory Factor Analysis.** For each item only factor loadings >.30 are reported, except for the five items that were excluded from Study 2, for which factor loadings >.10 are reported to show their ambiguity.

| Item | Factor 1 Behaviors | Factor 2 Cognitions | Factor 3 Emotions- Cognitions | Commonality |
|---|---|---|---|---|
| B3-I would report to the competent authorities illegal activities committed by a member of mafia organized crime, if I knew about them | 0.78 | | | 0.64 |
| B4-I would not rebel against a member of mafia organized crime making an extortion to me | 0.33 | | | 0.12 |
| B7-If I saw a person threatened by a member of mafia organized crime, I would report it to the competent authorities | 0.69 | | | 0.44 |
| B12-I would participate in events against mafia organized crime | 0.43 | | | 0.34 |
| B19-Iwould invest my money to organize initiatives against mafia organized crime | 0.39 | | | 0.19 |
| B20-I would not report a relative of mine if I knew that he/she collaborated with mafia organized crime | 0.39 | | | 0.26 |
| B21-I would report the illegal activities of mafia organized crime even if that would mean going against my neighbours and friends | 0.78 | | | 0.59 |
| E5-How much sadness mafia organized crime evoke in you? R | | | 0.45 | 0.18 |
| E7-How much anger mafia organized crime evoke in you? R | | | 0.61 | 0.43 |
| E9-How much fear mafia organized crime evoke in you? R | | | 0.40 | 0.15 |
| E11-How much indifference mafia organized crime evoke in you? | | | 0.36 | 0.24 |
| C1-Members of mafia organized crime are less honest than other people R | | | 0.51 | 0.23 |
| C6-Some values transmitted by mafia organized crime clans are agreeable | | | 0.34 | 0.24 |
| C10-What it is called mafia organized crime is actually common crime | | | 0.35 | 0.20 |
| C15-Mafia organized crime impoverishes the territories where it is present R | | | 0.35 | 0.14 |
| C30-Members of mafia organized crime think only about their own interest R | | | 0.31 | 0.12 |
| C8-Being affiliated to mafia organized crime open pave your way to richness | | 0.54 | | 0.28 |
| C13-Members of organized crime are good at their activities | | 0.49 | | 0.27 |
| C23-Members of mafia organized crime do just what is right for their clan | | 0.38 | | 0.15 |
| C25-Affiliating to mafia organized crime is a quick way to gain a lot of money | | 0.52 | | 0.27 |

**Table 1** (*continued*)

| Item | Factor 1 Behaviors | Factor 2 Cognitions | Factor 3 Emotions-Cognitions | Commonality |
|---|---|---|---|---|
| C29-A member of mafia organized crime is smarter than other people | | 0.58 | | 0.32 |
| C34-Members of mafia organized crime know how to be respected | | 0.56 | | 0.36 |
| C35-Mafia organized crime offers you job opportunities that the State is not able to give | | 0.40 | | 0.27 |
| C7-There are other problems in Italy that should have priority over mafia organized crime[*] | 0.27 | 0.21 | 0.25 | 0.27 |
| C18-It is not fair taking parental responsibility away from mafia organized crime members[*] | | 0.28 | 0.13 | 0.10 |
| C24-If the State would seek a dialogue with mafia organized crime, there would not be all this violence[*] | | 0.26 | 0.14 | 0.14 |
| C27-The important thing is that mafia organzized crime would not cause troubles to me[*] | 0.15 | 0.17 | 0.11 | 0.10 |
| C37-The State should deal with its corrupted politicians instead of thinking about mafia organized crime[*] | | 0.28 | 0.24 | 0.17 |

**Notes.**
[*]excluded items for Study 2.
R, reversed score.

or master's degree, 31 (7.8%) a doctoral degree or a master post-lauream, and 3 (0.8%) had other diplomas. On a scale ranging from 0 to 100 participants' mean self-reported socio-economic status was 44.67 (SD = 17.86). Seventy-two (18.3%) participants had an income <10,000 euros, 96 (24.4%) between 10,000 and 20,000, 99 (25.2%) between 20,000 and 30,000, 66 (16.8%) between 30,000 and 40,000, 24 (6.1%) between 40,000 and 50,000, and 36 (9.2%) >50,000.

### Measures

Attitude towards Italian Mafias Scale (AIMS): this scale, devised in light of the results from Study 1, consisted of 23 items to which participants were asked to express their disagreement or agreement on a response scale ranging from 1 (completely disagree) to 7 (completely agree).

Criminal Sentiment Scale (CSS, *Gendreau et al., 1979*), revised by *Shields & Simourd, 1991*, also present in *Simourd, 1997*; *Simourd & Olver, 2002*): this scale consisted of 41 items divided in five subscales each referring to an attitudinal object related to crime in general (*i.e.,* Law, Court, Police, Tolerance towards violations and Identification with criminal others). We chose to use the CSS for testing the discriminant validity of the AIMS because there is no existing measure that directly examines our specific construct of interest, *i.e.,* the attitudes towards Italian mafias. The CSS scale has been proven to be a reliable and valid measure for assessing attitudes towards crime, which is closely related to our construct but still distinct from it. Here, we used the revised version by *Shields & Simourd (1991)* where participants are asked to express their disagreement or agreement on a 3-point Likert scale ranging from 0 to 1. For consistency reasons, we used an adapted 7-point response scale ranging from 1 (completely disagree) to 7 (completely agree). To be sure that the Italian

version that we presented was comparable to the original one, we asked a native English speaker fluent in Italian to back-translate the Italian version into English. Since the two versions were almost identical, we could safely use the Italian version.

Donation: we offered participants the possibility to donate their compensation for completing the study to "Libera", a non-profit association active against mafias, corruption, and all forms of illegal activities (https://www.libera.it/schede-1326-libera_inglese). Participants could answer "yes" to donate or "no" to not donate and keep the compensation for themselves.

Demographic: we collected participants' age, gender, region of birth, education, income, and socio-economic status.

### *Procedure*

Participants were re-directed from Prolific to the online platform where we implemented the study (Qualtrics, Provo, UT). Once there, participants read the informed consent and clicked on a button to accept its terms. Then, they had to fill in the demographic questions, the AIMS, and the CSS. The order of presentation of the AIMS and CSS as well as the questions within the two questionnaires was random.

Finally, participants had to respond to the offer of donating their compensation of 1.50 euro to the non-profit association against Mafias.

At the end of the procedure, that took approximately 15 min, participants were debriefed and told that they would have been paid regardless their decision to donate.

The present research (Prot. n. 0002192) was approved by the Sapienza University of Rome ethics committee and was conducted in accordance with the 1964 Declaration of Helsinki.

## RESULTS

### Confirmatory factor analysis

We conducted a confirmatory factor analysis (CFA) on the 23-item of the AIMS by means of the R packages *lavaan* (*Rosseel, 2012*) and *semTools* (*Jorgensen et al., 2021*). Due to the ordinal nature of the response scale, we used the diagonally weighted least squares (DWLS) estimation method. No missing values were present in this study.

Given the results of the EFA in Study 1, which supported the theoretical basis of this research and represented by the tripartite model of attitudes, we performed the CFA on three main models, each assuming a structure with three components. However, to explore the possibility that the latent structure of the AIMS might still be univocal, we tested a fourth unidimensional model.

Thus, we built:

(1) a bifactor model, in which all the items loaded onto a general factor, and sub-sets of items loaded onto three grouping orthogonal factors;

(2) a second-order model, in which three correlated sub-sets of items were explained by a superordinate factor;

(3) a tridimensional model, in which sub-sets of items loaded onto three specific factors, but no general or superordinate factor was assumed;

**Table 2 Study 2 model comparison after confirmatory factor analysis.** For each model indexes of fitness are reported: chi-squared, degrees of freedom, comparative fit index, tucker-lewis index, standardized root mean square residual, and root mean square error of approximation.

| Model | $\chi 2$ | df | CFI | TLI | SRMR | RMSEA |
|---|---|---|---|---|---|---|
| Bifactor | 401.10 | 207.00 | 0.987 | 0.984 | 0.057 | 0.049 |
| Second-order | 507.767 | 132.000 | 0.971 | 0.966 | 0.073 | 0.085 |
| Tridimensional | 942.14 | 227.00 | 0.951 | 0.946 | 0.083 | 0.090 |
| Unidimensional | 2192.81 | 230.00 | 0.866 | 0.853 | 0.121 | 0.148 |

**Table 3 Study 2 Model comparison after Confirmatory Factor Analysis between bifactor models.** Comparative fitness index, tucker-lewis index, standardized root mean square residual, and root mean square error of approximation for the 18-item, 19-item and 23-item bifactor models are reported.

| Model | CFI | TLI | SRMR | RMSEA |
|---|---|---|---|---|
| 23-item | 0.987 | 0.984 | 0.057 | 0.049 |
| 19-item | 0.991 | 0.989 | 0.053 | 0.047 |
| 18-item | 0.991 | 0.989 | 0.053 | 0.049 |

(4) a unidimensional model in which all the items loaded onto one factor.

According to the standard criteria for the goodness of fit indexes (*Baumgartner & Homburg, 1996*; *Hu & Bentler, 1999*), the unidimensional model resulted in an insufficient fit. Conversely, the tridimensional and the second-order model had much better fits. Nonetheless, the best fit was achieved by the bifactor model (see Table 2). Model comparisons confirmed that the bifactor model ($\chi^2(227) = 401.10$) significantly outperformed the other models (tridimensional: $\chi^2 (207) = 942.15, p < .001$; second-order: $\chi^2 (207) = 942.15, p < .001$; unidimensional: $\chi^2(230) = 2192.81, p < .001$), indicating that a structure with a general factor (Mafia Attitude) and three specific factors (Behaviors, Cognitions, Emotions-Cognitions) was the most compatible with our data.

However, in terms of local fit, a closer inspection of the bifactor model revealed that some items loaded poorly on the general factor Mafia Attitude, which might signal a negligible contribution to the scale, if not possible misspecification issues. Thus, we decided to remove those items that loaded less than |.3| onto the general factor (E9, C8, C23 and C25) which resulted in a 19-item scale. A new inspection of the model showed that one further item (C10) resulted in a loading below the cutoff of |.3|, and for this reason was removed. The new 18-item bifactor model showed a better fit than the original 23-item model, but a slightly worse fit compared to the 19-item model (see Table 3). Nonetheless, because this model appeared simpler and in accordance with the criterium used for item removal (*i.e.,* loadings <|.3| into the general factor), we decided to retain the 18-item model for the following analyses (see Table 4 for items list and factor loadings).

We performed another model comparison between the aforementioned models relative to the new 18-item scale: again, the bifactor model($\chi^2(117) = 228.16$) significantly outperformed the other models (tridimensional: $\chi^2 (132) = 507.77, p < .001$; second-order: $\chi^2 (132) = 507.77, p < .001$; unidimensional: $\chi^2(135) = 1173.08, p < .001$).

**Table 4 Study 2 18-items bifactor model's factor loadings and commonality after Confirmatory Factor Analysis.** For each item factor loadings relative to the general factor and specific factors, and commonality are reported.

| Item | General Factor Mafia Attitude | Factor 1 Behaviors | Factor 2 Cognitions | Factor 3 Emotions-Cognitions | Commonality |
|---|---|---|---|---|---|
| B3-I would report to the competent authorities illegal activities committed by a member of mafia organized crime, if I knew about them | 0.54 | 0.69 | | | 0.78 |
| B4-I would not rebel against a member of mafia organized crime making an extortion to me | 0.42 | 0.34 | | | 0.29 |
| B7-If I saw a person threatened by a member of mafia organized crime, I would report it to the competent authorities | 0.55 | 0.68 | | | 0.76 |
| B12-I would participate in events against mafia organized crime | 0.71 | 0.08 | | | 0.51 |
| B19-Iwould invest my money to organize initiatives against mafia organized crime | 0.63 | 0.10 | | | 0.41 |
| B20-I would not report a relative of mine if I knew that he/she collaborated with mafia organized crime | 0.53 | 0.46 | | | 0.49 |
| B21-I would report the illegal activities of mafia organized crime even if that would mean going against my neighbours and friends | 0.53 | 0.65 | | | 0.70 |
| E5-How much sadness mafia organized crime evoke in you? R | 0.53 | | | 0.53 | 0.56 |
| E7-How much anger mafia organized crime evoke in you? R | 0.59 | | | 0.49 | 0.59 |
| E11-How much indifference mafia organized crime evoke in you? | 0.53 | | | 0.14 | 0.30 |
| C1-Members of mafia organized crime are less honest than other people R | 0.48 | | | 0.36 | 0.37 |
| C6-Some values transmitted by mafia organized crime clans are acceptable (agreeable?) | 0.63 | | | 0.18 | 0.43 |
| C15-Mafia organized crime impoverishes the territories where it is present R | 0.55 | | | 0.53 | 0.59 |
| C30-Members of mafia organized crime think only about their own interest R | 0.60 | | | 0.52 | 0.63 |
| C13-Members of organized crime are good at their activities (business?) | 0.37 | | 0.56 | | 0.45 |
| C29-A member of mafia organized crime is smarter than other people | 0.42 | | 0.36 | | 0.31 |
| C34-Members of mafia organized crime know how to be respected | 0.39 | | 0.6 | | 0.51 |
| C35-Mafia organized crime offers you job opportunities that the State is not able to give | 0.40 | | 0.36 | | 0.29 |

**Notes.**

R, reversed score.

**Table 5** **Study 2 18-item bifactor model internal consistency.** McDonald's Omega total ($\omega$T) and Omega hierarchical ($\omega$H) are reported for the general factor *Mafia Attitude* and for the three specific factor *Behaviors, Cognitions and Emotions-Cognitions*.

| Factor | $\omega$T | $\omega$H |
|---|---|---|
| *Mafia Attitude* | 0.92 | 0.78 |
| *Behavior* | 0.87 | 0.60 |
| *Cognitions* | 0.71 | 0.29 |
| *Emotions-Cognitions* | 0.88 | 0.59 |

**Table 6** **Study 2 Model comparison for construct validity.** For discriminant, convergent and unidimensional models chi-square, degrees of freedom, akaike information criterion, and bayesian information criterion are reported.

| Model | $\chi^2$ | df | AIC | BIC |
|---|---|---|---|---|
| *Discriminant* | 401.10 | 1592 | 75493 | 76200 |
| *Convergent* | 507.767 | 1593 | 75661 | 76364 |
| *Unidimensional* | 2192.81 | 1652 | 78234 | 78703 |

## Internal consistency

To measure the scale's internal consistency we computed the McDonald's $\omega$ (*Flora, 2020*) for the general factor and for the three specific factors of the bifactor model: as shown in Table 5 omegas total were good to excellent for the general factor Mafia Attitude and for Behaviors and Emotion-Cognitions subscales (*Nunnally, 1978*, see also *Nájera Catalán, 2019*), while still acceptable for Cognitions subscale. However, omega hierarchical indices demonstrate that a substantial proportion of scale internal consistency can be attributed to the general factor.

## Construct validity

To test the AIMS' discriminant validity we confronted participants' scores on the scale and their scores on the CSS. To do so, we performed a CFA on three different models:

(1) a discriminant model where we specified two macro-factors, one for the AIMS and one for the CSS with their subscales;

(2) a unidimensional model where we specified one macro-factor comprising the items of AIMS and CSS all together;

(3) a convergent model where we specified a macro-factor comprising both AIMS and CSS and their subscales.

To avoid under identification problems due to the high number of parameters to be estimated compared to the number of observations, we used a robust maximum likelihood (MLR) estimation method. Model comparison revealed that the discriminant model outperformed the other two, giving evidence for discriminant validity (see Table 6).

To furtherly support this claim we performed a correlational analysis between participants' scores on the AIMS and their scores on the CSS. To do so, we first created an index for the AIMS by averaging its items, such that higher scores indicated pro-mafia attitudes. We then created an index also for the CSS by following the procedure described
in *Shields & Simourd (1991)*. To increase its interpretability, we reversed the CSS scores such that higher values indicated pro-crime attitudes.

We performed a Spearman's rank-order correlation for non-parametric tests between the two questionnaires: results showed a positive and significant correlation (rho = .45, $p < .001$), indicating that a positive attitude towards mafias corresponded to a positive attitude towards crime. The moderate value of this correlation, combined to the results of the CFA, gave evidence for discriminant validity of the AIMS. However, as a final proof, we calculated the covariance score between the AIMS and the CSS obtained from the Discriminant model, which yielded a r = .40, 95% CI [.26, .54], well below the suggested value of .8, above which there might be discriminant validity problems (*Rönkkö & Cho, 2022*).

### Criterion validity - predictive validity

We performed a logistic regression using the R package *car* (*Fox & Weisberg, 2019*) with Donation as our dichotomic dependent variable and AIMS score as our predictor.

The model ($\chi^2(1)$ =18.98, $p < .001$) showed that AIMS did predict donation behavior ($b = 0.58$, SE = 0.14, $z = 4.20$, $p <. 001$). Specifically, AIMS scores 1 sd below the mean corresponded to 62% probability of donating, while AIMS score 1 sd above the mean corresponded to only 40%, suggesting that pro-mafia attitudes corresponded to a lower probability of donating to an association against mafias.

Crucially, to discriminate the predictive validity of the AIMS from the CSS, we performed a logistic multiple regression with Donation as our dichotomic dependent variable and AIMS and CSS scores as our predictors. Results indicated that - when CSS was held constant- the AIMS still predicted whether participants donated their reward ($b = -0.60$, SE = 0.15, $z = 3.92$, $p <. 001$), but this did not hold true for the CSS ($b = -0.06$, SE = 0.15, $z = -0.41$, $p = .68$).

### Measurement invariance

Since experiences with mafias vary across different territories, it is reasonable to assume that individuals from regions where the presence of mafias is more tangible may have a different social representation of the phenomenon compared to those from areas where mafias are less pervasive. Indeed, mafias are influenced by cultural and social elements that are specific to the regions where they originated (see *Travaglino et al., 2014*; *Travaglino, Abrams & De Moura, 2016*). In this vein, certain situations described in the scale items may be perceived differently in different regions, potentially impacting the structure of the scale itself. Thus, to deal with this issue, we tested measurement invariance between participants from the five DRMR (*i.e.,* Apulia, Basilicata, Calabria, Campania and Sicily) and those from the rest of Italy. Due to the high number of parameters to be estimated and since some response categories in some items had no observations, we merged 6 and seven response categories into the same category to obtain a 6-point response scale (instead of the original 7-point response scale). Although treating ordinal data as continuous can be an acceptable alternative when having more than five response categories (*Distefano & Morgan, 2014*), at this stage we decided to respect its original nature and collapse response

categories, which seems a more valid approach when dealing with sparse data or small frequency cells (*Distefano, Shi & Morgan, 2021*). We then performed a multi-group CFA using the diagonally weighted least squares (DWLS) estimation method. Following *Svetina, Rutkowski & Rutkowski (2019)* procedure for categorical variables, we started by testing the configural invariance by fitting a baseline unrestricted model, which showed a very good fit ($\chi^2 = 293.45$, $df = 234.00$, comparative fit index [CFI] = 0.995, Tucker-Lewis index [TLI] = 0.994, standardized root mean square residual [SRMR] = 0.063, root mean square error of approximation [RMSEA] = 0.036). Then, we fitted a model where we forced the thresholds to be equal in the two groups, and this did not significantly decrease the goodness of fit compared to the baseline model ($\chi^2$ baseline = 293.45, $\chi^2$ thresholds = 312.13; $p = .63$). Finally, we fitted a model where both thresholds and loadings were set to be equal between the two groups and compare its fitness to the thresholds only model. Again, results showed no significant decrease in goodness of fit ($\chi^2$ thresholds = 312.13, $\chi^2$ loadings = 386.71; $p = .08$), which confirmed that the AIMS measured the latent construct in the same way among participants from the five DRMR and those from the rest of Italy.

## Comparing AIMS scores between deep-rooted mafia regions and rest of Italy

Once established the measurement invariance, we explored if participants from the five DRMR and those from the rest of Italy differ in their AIMS scores. Indeed, parallel to the purported differences in item perception that could impact the structure of the scale, participants' attitudes may depend on their familiarity with contexts characterized by mafia's pervasive and tangible presence (as in the DRMR) or contexts where this presence is comparatively more subtle or even imperceptible.

We used JASP (Version 0.16.3; *JASP Team, 2022*) to perform a *t*-test on participants' mean scores (AIMS means scores), obtained by the widely used procedure of averaging the responses to the items of the scale. Here we reported the analyses and the results on the scores derived from the model built for testing measurement invariance, where we reduced the response categories from the initial seven to six. We performed the same analyses also on the scores based on seven response categories, yielding the same pattern of results (see Supplementary materials).

Moreover, given the exploratory nature of these analyses, we applied both a frequentist and a Bayesian approach to test the robustness of the results and to provide a quantification of the evidence in favor of the hypothesis that there would be a difference between the two samples over the hypothesis that there would not be any difference.

Finally, to strengthen our results, we performed the same analysis also on participants' latent general factor scores (AIMS fscores) obtained from the CFA, a method that weighs the contribution of each item to the scale. Nonetheless, it has been observed that, if the scale tends toward unidimensionality (such as our bifactor solution), using averaged (or summed) scores might be even better. Indeed, when the scale is unidimensional these scores are less sensitive to specific characteristics of the sample recruited (*Widaman & Revelle, 2023*). Analyses of the AIMS fscores are reported in the Supplementary materials.

### Frequentist approach

Since it is common practice to treat aggregated scores deriving from an ordinal Likert scale with five or more response categories as continuous and perform parametric analyses (*e.g.,* *Norman, 2010*), we performed a parametric $t$-test on AIMS scores.

Results on AIMS scores revealed a significant difference t(391) $= -2.14$, $p = .03$, Cohen's d $= -.22$, 95% CI [$-.41, -.02$]), with more negative attitudes towards mafias expressed by the DRMR participants ($M_{DRMR} = 2.43$) compared to those from the rest of Italy ($M_{Rest} = 2.60$).

### Bayesian approach

We performed two Bayesian independent sample t-tests on the same variables with a Cauchy prior distribution set at the default scale value of 0.707. Contrary to the results obtained with the frequentist approach, the analysis on AIMS mean scores suggested that we do not have enough evidence ($BF_{10} = 1.02$) to support a difference between DRMR participants ($M_{DRMR} = 2.43$, 95% CI [2.32, 2.54]) and those from the rest of Italy ($M_{Rest} = 2.60$, 95% CI [2.49, 2.70]). The error percentage was equal to 0.02%, which indicated a good stability in the algorithm that was used in this analysis. Moreover, across a wide range of widths the Bayes factor appeared to be relatively stable, ranging from 0.55 to 2.08.

Finally, we tested if the two groups differ for other relevant demographic characteristics: we found no differences in terms of age (t(388.86) $= 1.21$, $p = .22$; $M_{DRMR} = 28.24$, $M_{Rest} = 27.23$), socio-economic status (t(384.16) $= -0.73$, $p = .46$; $M_{DRMR} = 44.02$, $M_{Rest} = 45.33$) or education ($\chi^2(4) = 5.70$, $p = .22$). Conversely, they differ for their income ($\chi^2(5) = 19.76$, $p = .001$), with participants from the rest of Italy being richer than those from the five DRMR regions.

## DISCUSSION

Italian mafias continue to be an open issue for their territories of origin as well as for the rest of Italy and for many Countries worldwide. Pro-mafias attitudes might explain the support that allow these criminal associations to infiltrate and proliferate in many societies. However, despite their importance, psychometrically reliable instruments to measure attitudes towards mafias are lacking. Thus, to fill this gap we developed the Attitudes towards Italian Mafias Scale (AIMS). Following exploratory and confirmatory factor analyses we were able to create an 18-item questionnaire whose latent structure was best identified by a bifactor model, with a general factor Mafia Attitude, and three more specific orthogonal factors, namely, Behaviors, Cognitions and Emotions-Cognitions.

Bifactor models, which are special cases of second-order models, are not often used in psychological research, but can be very useful as they allow to estimate a unidimensional construct while acknowledging its multidimensionality (*Reise, Moore & Haviland, 2010* in *Boateng et al., 2018*). Furthermore, these models allow for partitioning the variance of each item between the variance explained by the general factor and the variance explained by the specific factors, which can be seen as different manifestations of the construct of interest. In our case, the bifactor model showed an excellent goodness of fit and clearly outperformed the unidimensional and second-order/tridimensional models,

demonstrating that the explicit attitude towards Italian mafias is neither univocal nor a construct consisting of separated components or nuisances. In fact, behaviors, cognitions and emotions appear as actual diverse manifestations of a unitary psychological construct (*Reise, Bonifay & Haviland, 2018*). Interestingly, within the three manifestations, the factor *Emotions-Cognitions* showed a mixed-nature, with some items more related to the emotions domain and some items related to the cognitions one. This result would confirm that the three manifestations of attitude do not always appear sharply separated (*Eagly & Chaiken, 1993*), thus, it is even more important the presence of a strong general underlying factor such as that identified in our analysis.

In a similar vein, the scale shows a good to excellent internal consistency for the general factor Mafia Attitude as well as for Behaviors and Emotions-Cognitions factors, and acceptable for Cognitions factor. However, hierarchical omega values indicates that the general factor Mafia Attitude is the major responsible of the scale internal consistency.

By comparing the AIMS with the Criminal Sentiment Scale (CSS, *Gendreau et al., 1979*; *Shields & Simourd, 1991*), a questionnaire extensively used to investigate attitudes towards crime in general, we also demonstrated its discriminant validity. The two scales are in fact related but moderately (rho = .45). Moreover, a CFA showed that the model that best describes the relationship between the two scales is a model where AIMS and CSS are treated as separate and distinct macro-factors: mafia-like organizations are, indeed, related to crime, but, because of their cultural, social and political connotation (*Allum, Merlino & Colletti, 2019*), they can be ascribed to a phenomenon representing a specific attitudinal object. This claim was further supported by the covariance value between the AIMS and the CSS obtained from the Discriminant model, which with a r = .40, 95% CI [.26, .54], fell well below the suggested value of .8 (*Rönkkö & Cho, 2022*), above which there might be discriminant validity problems.

Crucially, the increased tendency to donate to the non-profit association showed by those participants with stronger anti-mafias attitudes (or weaker pro-mafia attitudes) demonstrates the predictive validity of the AIMS. Although not always strong and straightforward (*Wicker, 1969*), attitudes are often reliable predictors of many actual behaviors (*Ajzen & Fishbein, 1977*), especially when attitudes and behaviors are measured at corresponding levels of specificity (*Kraus, 1995*). In fact, from a semantic point of view, the donation to an association against mafias is strongly related to the item "I would invest some money to organize initiatives against mafia organized crime" listed in the specific behavioral factor of the AIMS. As further confirmation, behavioral intentions are the key component in the relationship between attitudes and behaviors (*Ajzen, 2005*).

Finally, the AIMS was invariant to participants' region of birth, indicating that the scale was interpreted in the same manner by people from the five deep-rooted mafia regions (DRMR, *i.e.,* Apulia, Basilicata, Calabria, Campania and Sicily) and people from the other parts of Italy. This allowed us to explore the difference in the AIMS scores between the two groups of participants. Despite the low scores indicated overall negative attitudes towards mafias, across robust frequentist and Bayesian analyses we found that those expressed by DRMR participants were more negative compared to those of the participants from the rest of Italy. A two-facet explanation can be offered to explain this, somewhat unexpected,

pattern of results. On the one hand, it can be posited that after decades of silence code and even overt ideological support, participants from the DRMR are finally reacting leading to a 'redemption' process that can re-balance the difference with the other Italian regions. Even if somewhat speculative, this interpretation is supported by the spread of associations and initiatives aimed at contrasting mafia-like organizations (*Cinotti, 2015*). On the other hand, the less positive attitudes expressed by the participants born in the five DRMR, may reflect the fact that in these territories mafias often adopt a non-violent strategy that aims at gaining the largest capitals possible (*Giorgi et al., 2018*). In this vein, mafias can be even a means to get rich and, thus, something to make deals with rather than to fight against (*Giorgi et al., 2018*).

It is worth also noting that, although the people from the five DRMR regions were significantly poorer than those of the rest of Italy, both groups were quite well educated. We, thus, underline the importance of having the adequate instruments to understand (anti)social phenomena in order to reject them despite one's own socio-economic conditions.

## LIMITATIONS AND FUTURE DIRECTIONS

One limitation of this research regards the use of convenience samples, which, especially for sensitive topics such as the one here investigated, might suffer from the well-documented selection bias (*Heckman, 1990*), and might fail to represent the general population (*Paolacci, Chandler & Ipeirotis, 2010*). This could be especially accurate for Study 1, wherein we enlisted a more imbalanced population, particularly regarding their birth region and their education. However, while in Study 2 we implemented a systematic plan that necessitated a specific sampling approach, Study 1 was designed as exploratory, without any specific predictions or hypotheses that would have required specific sampling constraints. In this vein, we did not have any specific expectations or hypotheses regarding in particular the impact of participants' education level on the perception of the items and, consequently, on the structure of the scale. We reasoned that as long as the items are comprehensible their perception should remain unaffected. Therefore, during the item preparation stage, we took special care to ensure that the content and format of the items were understandable by consulting various experts in the field.

Moreover, 59.6% and 56.5% of our participants in Study 1 and Study 2, respectively, reported having at least a bachelor's degree, that closely aligns with what emerges from a report produced by the National Institute of Statistics (ISTAT) in 2022, showing that 63% of Italian people aged between 25 and 64 years hold an academic degree, with a 0.3% increase compared to 2021. It is however essential to acknowledge that this data overlooks an important segment of the Italian population comprising individuals over 65 years old. While this is true, it is also worth considering that the age range of 24–65 years old includes a significant part of the productive population, with whom the Mafia is likely to interact and conduct business. Consequently, for future interventions and educational programs, it may be more beneficial to survey and target this specific segment of the population. With this in mind, the imbalance of our sample in terms of educational level might not be an alarming issue.

Another important advantage of these participants is that they usually are more heterogeneous than standard Internet samples (*Berinsky, Huber & Lenz, 2012*; *Buhrmester, Kwang & Gosling, 2011*). Moreover, since they often time have higher motivation due to either their relationship with the experimenters, their interest in the topic or the presence of external incentives (*Jun, Hsieh & Reinecke, 2017*), these participants overall provide good quality and reliable data (*Stanton et al., 2022*). Future research might test the AIMS in other contexts and populations, for instance convicted mafia criminals (*Craparo et al., 2018*; *Salvato et al., 2020*). As a further step, we hope the AIMS might evolve to measure attitudes towards mafia-like organizations originating from other parts of the World, either present in Italy itself, such as the Nigerian and Albanian mafias, or present in other countries, such as the powerful Russian or Japanese mafias. In this regard, although these organizations share with the Italian mafias their modus operandi, being they a product of cultural, social, and political elements, they have contextual idiosyncrasies that require ad hoc changes to the AIMS.

Another potential limitation lies in the intrinsic nature of self-report measures: it has been known for decades now that respondents might not be particularly aware of their higher order mental processes, such as the decision-making driving their choices (*Nisbett et al., 1977*). This lack of introspection might lead to biased responses that do not reflect people's real thoughts or judgements. However, when stimuli are influential and salient, people might be more aware of their internal mental states and provide more accurate responses (*Nisbett et al., 1977*). In this vein, mafias are an extremely relevant issue for Italian (and not only) society, which should elicit the necessary involvement to provide accurate responses.

The saliency but at the same time sensitivity of the topic itself, though, might be responsible of the second important limitation, that is, when providing self-reported answers people tend to respond in a socially desirable manner to convey a positive self-image (*Edwards, 1953*; *Krumpal, 2013*). In fact, social desirability refers to either a personality trait identified by a constant need of approval, but also to the characteristics of the stimulus (*i.e.,* item), whereby, if the response to that stimulus violates the social norm, people tend to underreport the socially undesirable behavior, but if the response conforms to the norm, then people tend to overreport the socially desirable behavior (*Fowler Jr & Fowler, 1995* in *Krumpal, 2013*). The AIMS require answers both violating and conforming to social norms, which might elicit social desirability. However, the methods used for collecting and analyzing data allowed us to assure participants anonymity, which, by reducing perceived risk and losses to respond in a socially undesirable manner, is one of the main strategy employed to incentivize truthful responses (*Rasinski et al., 1999* in *Krumpal, 2013*).

It is worth noting that participants were able to donate only a small amount of money to the anti-mafia association Libera. While this limited donation might not have been particularly significant for them, we believe that this limitation does not undermine the validity of our results or our conclusions. In fact, even though one could easily dismiss such a small amount of money and agree to donate it, a person with a positive attitude towards Italian mafias would be unlikely to contribute their earnings to an association that likely

holds opposing ideological values. If anything, with a larger reward, we would expect to see an even stronger polarization, with those who express more negative attitudes towards Italian mafias more likely to donate to anti-mafia associations compared to those with more positive attitudes.

Lastly, while the fit indexes throughout the analyses confirmed the appropriateness of our choices, it is important to note that the 3-factor solution derived from the exploratory factor analysis (EFA) in Study 1 explained only 26% of the total variance. This percentage falls below the commonly accepted threshold typically applied to behavioral data, which is between 50% and 60% (*Peterson, 2000*). However, this threshold is a matter of debate as seldom it has theoretically and empirically justified (*Peterson, 2000*). In fact, this rule of thumb often fails to take into account the specific characteristics of the research context and, more importantly, of the research object. In view of this, we believe that the low value observed may be attributed to the inherent complexity of the psychological construct itself. Indeed, attitudes towards crime-related phenomena, such as Italian mafias, encompass numerous and diverse aspects, some of which may be substantially distinct from each other.

## CONCLUSIONS

Italian mafias, with their criminal activities, represent a relevant phenomenon causing severe social and economic consequences. Psycho-social research has demonstrated the usefulness of attitudes for explaining and changing socially undesirable behaviors. Nonetheless, research lacked a psychometrically valid and reliable tool for investigating attitudes towards Italian mafias. To fill this gap we developed the AIMS, an 18-item questionnaire that measures the behavioral intentions, thoughts and emotions associated to the Italian mafias. The scale might help researchers and professionals to survey the sentiment of the population and promote initiatives and interventions aimed at deconstructing supportive tendencies towards mafia-like organizations.

## ACKNOWLEDGEMENTS

We thank Patrizia Torretta and Cristina Bonucchi for their help in revising the content of the items of the scale. We thank Emanuele Polizza for his help in designing the study, translating the items, and collecting the data. We also thank Anita Anderson for translating some of the materials. We thank all the people involved in the ''Progetti Operativi Nazionali'' (PON) for legality, the Italian Postal Police and the Italian Ministry of the Interior for supporting the project.

### Funding

This research was supported by the small grant ''Avvio alla Ricerca'' tipo II (n. AR220172B7D7C570) from Sapienza University of Rome and awarded to Michael Schepisi for participants payment, and by the grant given by Ministero dell'Interno

(CUP F23J18000050007) within the Project: "Nei Panni di Caino capire e difendere le ragioni di Abele (Educazione alla legalitá per la prevenzione di comportamenti antisociali espressi anche attraverso l'uso della realtá virtuale immersiva)" PON Legalitá FESR/FSE 2014-2020 for APC payment. The funders had no role in study design, data collection and analysis, decision to publish, or preparation of the manuscript.

## Grant Disclosures

The following grant information was disclosed by the authors:
"Avvio alla Ricerca" tipo II: AR220172B7D7C570.
Ministero dell'Interno: CUP F23J18000050007.

## Competing Interests

Marco Tullio Liuzza is an Academic Editor for PeerJ.

## Author Contributions

- Michael Schepisi conceived and designed the experiments, performed the experiments, analyzed the data, prepared figures and/or tables, authored or reviewed drafts of the article, and approved the final draft.
- Marco Tullio Liuzza conceived and designed the experiments, performed the experiments, analyzed the data, authored or reviewed drafts of the article, and approved the final draft.
- Althea Frisanco conceived and designed the experiments, performed the experiments, authored or reviewed drafts of the article, and approved the final draft.
- Anna Maria Giannini conceived and designed the experiments, authored or reviewed drafts of the article, and approved the final draft.
- Salvatore Maria Aglioti conceived and designed the experiments, authored or reviewed drafts of the article, and approved the final draft.

## Human Ethics

The following information was supplied relating to ethical approvals (*i.e.*, approving body and any reference numbers):

The present research (Prot. n. 0002192) was approved by the Sapienza University of Rome ethics committee and was conducted in accordance with the 1964 Declaration of Helsinki.

## Data Availability

The dataset and script are available at Mendeley: Schepisi, Michael; Liuzza, Marco Tullio; Frisanco, Althea; Giannini, Anna Maria; Aglioti, Salvatore Maria (2023), "Development and validation of the Attitudes towards Italian Mafias Scale (AIMS)", Mendeley Data, V2, doi: 10.17632/czpd76psmp.2.

## Supplemental Information

Supplemental information for this article can be found online at http://dx.doi.org/10.7717/peerj.16120#supplemental-information.

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
