# Peer review of "Attitudes towards Italian Mafias Scale (AIMS): development and validation"

_PeerJ, doi:10.7717/peerj.16120_

## Round 0.1 · original submission · Major Revisions

As you can see both Reviewers evaluated positively your paper, but they also proposed several suggestions and changes that should be addressed. In particular, both Reviewers asked for some clarifications both at the theoretical and methodological levels. The Reviewers proposed precise and detailed comments and I would ask you to consider all of them when revising your paper. Please let me know if you need more time to complete your revision (35 days is the standard deadline for this journal).

Reviewer 1 ·

Basic reporting

Dear Editors and Authors, thanks for providing me with the opportunity to review “Development and validation of the Attitudes towards Italian Mafias Scale (AIMS)”, which I read with interest and attention.
The paper aims to develop a scale for measuring attitudes towards Italian mafias, which is called Attitudes towards Italian Mafias (AIMS). The study is conducted in two parts, with the first part involving 292 participants and the second part involving 393 participants. In the first study, an exploratory factor analysis is used to reduce the number of items and identify latent constructs, while in the second study, a confirmatory factor analysis is used to validate the resulting 18-item questionnaire.
The analyses lead the Authors to assert that the AIMS has good discriminant validity compared to a measure of attitudes towards crime, predictive validity of donation behavior to an association against Mafias, internal consistency, and invariance for people of the five deep-rooted mafia regions of Southern Italy and those from the rest of Italy. The study also shows that there is a difference between the attitudes of people from the five deep-rooted mafia regions and those from the rest of Italy, with the former having more negative attitudes towards Mafias compared to the latter.
Overall, the paper provides a comprehensive description of the development of the AIMS scale and its validation using multiple methods. The findings suggest that the scale is a reliable and valid measure of attitudes towards Italian mafias and can be used in future research to examine the relationship between attitudes and behavior towards these criminal organizations.
The paper has a certain degree of originality and the AIMS might indeed be useful for further studies. Yet, the Authors should amend their study in several way in order to reach publication standards. In particular, I suggest the authors to:
• Edit their text to increase its readability: streamline certain sections, reshape and shorten hard to read sentences, reduce the use of references in the middle of their sentences, proofread the document, reduce the use of acronyms.
• Ameliorate their review of the literature on mafia which, at the moment, is often imprecise and a little superficial.
• State from the beginning the added values of their study. They should explain why it is relevant to develop a new metric to assess attitudes toward mafias and in which ways AIMS outperform other indicators.
• Convince the reader that 1) the lack of representativeness of their convenience sample does not bias too much the results of the stud; 2) a small donation to anti-mafia organizations such as Libera are capable of providing a measure of people’s attitude toward mafias.
In what it follows, I provide further details concerning some of these main issues.

Clear and unambiguous, professional English used throughout.
The article is written in English, but it does not use always clear text. The article conforms to professional standards of courtesy and expression.
The article is not effective in demonstrating how the work fits into the broader field of knowledge. In particular, it might be beneficial to discuss the existence of other indicators of attitudes toward organized crime.
The presentation of the literature on mafias might be ameliorated. See below:
The Authors might want to reshape the first paragraph of their introduction. As first, the Authors present the very broad definition of Organised Crime adopted by the UN; then, they present the distinctive traits of more structured criminal groups (Finckenauer, 2005); finally, they refer to Italian mafias a further subcategory. As currently presented, the reference to the UNs’ definition is misleading; the same Lavorgna and Sergi specify that the UNs’ definition is a sort of minimum threshold, which barely depict the mafia phenomenon. Paoli and Vander Beken 2014 reason on the nature of these definition in an effective manner.
“[…] Basilicata, no specific mafia-like organization has developed (apart from a clan called Basilischi).” The Authors might want to use the term organization or network of clans, etc. as Basilischi does not refer to a single clan (see Direzione Antimafia e Governo Italiano, Relazione Semestrale Antimafia or Sergi).
“Mafias have been responsible for the 9% of the homicides committed over the national territory”, about half of non-domestic/non-intimate homicides and more than 70% of the homicides related to systemic violence (see Aziani 2022).
“[…] two of the deep-rooted mafia regions showed a 16% GDP pro capite decrease (Pinotti, 2015).” Authors might want to rephrase this as Pinotti states that “The comparison of actual and counterfactual development shows that the presence of mafia lowers GDP per capita by 16%”, which is different.
“[…] Crucially, […] all over the World” see Calderoni, Berlusconi, et al. 2016.
Overall, the introductive section presents Italian mafia; in so doing, it briefly refers to the concept of the population support toward mafia groups. The Authors might want to expand on the aims and purposes of their study.
“Mafia is a typical Italian phenomenon, which needs to be dealt with primarily within its context of origin.” This statement is partially inconsistent with the sample of participants in study 1 (i.e., Fifty-five participants (18.8%) were born in Apulia, Basilicata, Calabria, Campania and Sicily).
Perhaps, the Authors might want to streamline section “Attitudes and their measures towards Mafia, criminal organizations and crime” by shortening the presentation of the limits of the approaches they are not going to use in their research.

Experimental design

The paper mostly presents a methodology to assess the attitudes toward mafias; as it is currently shaped, it does formulate neither a clear research question nor an hypothesis. Eventually, the Authors might decide to introduce them and then use the mehtod they present to answer to them. Nonetheless, their study might have value also as a more strictly methodological paper. In any case, the following poitns should be adressed:

“we eliminated three items (B6, C31, C33) because they correlated less than |.2| with the total 270 score of the questionnaire.” Is |.2| the correlation coefficient? The p-value of the significance test? Is the correlation significant? Is it correct to remove questions whose response does not align with the overall score? Perhaps they capture important components, which are not covered by other questions.
“3-factor solution explaining 26% of the total variance.” Perhaps the authors want to specify why they are satisfied with this.
“Criminal Sentiment Scale (CSS, Gendreau et al., 1979, revised by Shileds & Simour, 1991, also present in Simourd 1997, and Simourd & Olver, 2002): given the absence in the literature of a measure investigating the same construct of interest (i.e., the attitudes towards Italian Mafias), we presented participants with this widely used scale that investigates a similar but distinct construct, that is, the attitudes towards crime.” Rephrase.

Validity of the findings

As said, it sould be crucial to convince the reader that 1) the lack of representativeness of their convenience sample does not bias too much the results of the stud; 2) a small donation to anti-mafia organizations such as Libera are capable of providing a measure of people’s attitude toward mafias.
With respect to point 1, it might be worth to move upfront the discussion of the limitations (or a part of it) and to expand the discussion on passages as the follwoing: “Three respondents (1%) had a middle school diploma, 60 (20.5%) a 231 high school diploma, 174 (59.6%) a bachelor or masterís degree, 36 (12.3%) a doctoral degree or a master post-lauream, and 6 (2.1%) had other diplomas”. Perhaps this deserve further comments as this distribution seem to be very much different from the Italian population.
As for point 2, is it perhaps relevant to acknowledge that the monetary donation is particularly tiny (upward bias in the attitude against mafias) and that, over time, Libera has been strognly criticized becuase of its (alegged?) economic interests in the fight against mafias (downward bias in the attitude against mafias).

Additional comments

I do not have further comments.
Thanks again for involving me in this review process.

Best regards

·

Basic reporting

1 Basic reporting

1.1 English. It is hard for me to evaluate the quality of the English writing of the authors since I am not an English mother tongue speaker. What I can say is that I read the entire manuscript without any difficulties and everything seemed absolutely clear to me.

1.2 Intro and Background. The specific features of the "Italian mafia” is well described in the manuscript as well as it is clearly justified the need for a specific tool for assessing the attitude toward the criminal organizations that can be assimilated into the Italian Mafia.

1.3 Structure. The organization of paragraphs follows a logical structure and results in a remarkable clarity of the manuscript.

1.4 Figures. There are no figures in the manuscript, all the results are presented in tables. All the tables and captions are presented in a standard format and are easily readable.

1.5 Data. All collected data are accessible to reviewers in a comma-separated format and I’ve been able to download them without any problem. R scripts are accessible as well and they appear to be well commented.

Experimental design

2 Experimental design

2.1 Originality. To my knowledge, no specific scale assessing the attitudes toward the mafia has been proposed in the literature.

2.2 Meaningfulness. There are a lot of reasons why a psychometric instrument assessing the attitude toward the mafia is needed. I will briefly mention only the necessity to survey public opinion whenever actions contrasting organized crime are planned. This paper offers a short and reliable scale that can be easily adapted outside Italy in all the countries in which crime organizations share distinctive features with the Italian mafia

2.3, 2.4 Method. Data collection has been conducted in a standard way: spreading the link to an online questionnaire through social media. This procedure does not guarantee a representative sample of the population (see https://docs.iza.org/dp11799.pdf) but the aim of the two studies presented in the manuscript was not a population survey but, instead, to build a Likert scale assessing a widely shared attitude. In this regard the sampling procedure is correct.
Data collection and data analysis are presented in such a clear way that replicating the analysis is easy (I did it myself since both data and R-scripts are available).
Given the approach used by the authors, item generation and scale construction are essentially correct.

Validity of the findings

3 Results

3.1 Impact and Novelty. As I already mentioned, I am not aware of an alternative scale assessing the attitude toward the mafia. For this reason, this manuscript has to be published. There are some issues to pay more attention to and certainly future administrations of the scale, testing different samples, will contribute to a clearer assessment of the psychometric properties of the instrument.

3.2 Reliability. In the 2 studies reliability is essentially conceived as internal consistency of the scale and it is estimated via the McDonald’s ω index. It would be interesting in future administrations of the scale to assess also stability of the measure with a test-retest procedure.

3.3 Validity. The validity of the scale is essentially tested in two different ways: 1) discriminating the responses to the “attitude toward Italian mafias scale” (AIMS) and the responses to the Crime Sentiment Scale (CSS) and 2) predicting the intention to donate the participation reward to an Anti mafia association from the attitude scores. An index of Spearman rank-order correlation of .45 is quite large. The two scales have a significant part of the variance in common. The discriminating power of the model comparison should be taken with caution. An additional problem is that the conceptual relation between AIMS and CSS is not symmetrical (but the Spearman rank-order correlation is). All those who have a positive attitude toward the Italian mafia are expected to also have a positive “crime sentiment” but the reverse does not hold true. I suggest testing the hypothesis of a differential power of the two scores in predicting the intention to donate the participation reward.
The authors compute the predictive validity of the AIMS two times: one using the scale scores and the other using the latent factor scores. In my humble opinion, the second analysis is redundant and can be eliminated from the manuscript.

The analysis of the invariance of the measure in what the authors call “deep-rooted mafia regions” (Sicilia, Campania, Calabria e Puglia, to which the authors add Basilicata without theoretical reasons, probably only for geographical contiguity) is not opportunely theoretically justified. Why is it necessary to exclude that the attitude toward the Italian mafia is differentially organized in people living in regions that gave birth to mafia-like criminal organizations (mafia, camorra, ‘ndrangheta, sacra corona unita)?

The authors compared the attitude scores obtained from respondents living in the deep-rooted mafia regions with the score obtained from respondents living in the rest of Italy. Here too the reason for this comparison needs to be more theoretically justified. The results of the comparison show an unexpected result: those who live in a region that gave birth to a mafia-like criminal organization exhibit a lower attitude score compared to respondents living elsewhere. Any explanation for this result? This finding is commented in the lines 710 to 715 in a very short passage.
The statistical analysis bringing to this result are replicated using a Bayesian approach. The results of the Bayesian comparison is not always in line with the classical t-tests. I would have preferred a different approach: first justify theoretically what is expected from this comparison then choose a line of testing either classical statistic or Bayesian approach but not both of them.

Additional comments

With this simple correction the manuscript will be ready to be published. I am always available for a reading of a new version of this paper.

---

## Round 0.2 · Minor Revisions

I have now received the comments from the two Reviewers, who I want to thank again for their time and efforts in revising this work. As you can see, while one Reviewer is happy with this version, the other Reviewer proposed some small additional changes that can be addressed with a minor revision.

Reviewer 1 ·

Basic reporting

Dear Editor and dear Authors,
I read the revisions of “Development and validation of the Attitudes towards Italian Mafias Scale (AIMS)”. The Authors took into consideration all recommendations emerged during the first round of reviews. In most of cases, they succeed in amend and ameliorate the paper. In the few occasions in which they did not intervene on the text, their answers to my comments were convincing.
Aside carefully checking for typos (e.g., “60.000 associates” should be “60,000 associates” or first sentence of the abstract: “Attitudes towards Italian Mafias (AIMS)”, should Scale be there?), I have only one lasting minor comment regarding the “validity of the findings”.

Regards,

Experimental design

no comment

Validity of the findings

Authors comment on the fact that their sample for Study 1 is not representative of the overall population stating that “In this vein, we did not have any specific expectations or hypotheses on the impact of education on the perception of the items and, consequently, on the structure of the scale. As long as the items are comprehensible, their perception should remain unaffected. Therefore, during the item preparation stage, we took special care to ensure that the content and format of the items were understandable by consulting various experts in the field.” The consideration of participants' ability to fully comprehend the questionnaire addresses only one of the potential sources of bias related to using a non-representative sample in terms of education level. Another source of bias might arise from the differing attitudes toward mafias based on participants' educational backgrounds. It would be beneficial for the author to provide additional explanation on this matter, potentially by citing relevant literature that supports their methodological approach.

Additional comments

Finally, the title might be changed in: ‘Attitudes towards Italian Mafias Scale (AIMS): Development and validation’.

·

Basic reporting

I made no suggestion to the authors about topics in this section in the previous version of the manuscript

Experimental design

I made no suggestion to the authors about topics in this section in the previous version of the manuscript

Validity of the findings

all (but one) of the suggestions to the authors have been accepted and the manuscript has been revised consequently.
I suggested the authors to choose between a classic and a Bayesian approach to some of the analysis conducted but they opportunely justified the choice of reporting the results of both approaches

Additional comments

the second version of the manuscript is an excellent paper and it worth being published

---

## Round 0.3 · accepted · Accept

The authors have addressed all the reviewers' comments; I am happy with the current version.